# Synchronous motion of the Easter mantle plume and the East Pacific Rise

John M. O'Connor [1,2] ✉, Marcel Regelous[1], Karsten M. Haase [1], Christophe Hemond[3], Anthony A. P. Koppers [4], Daniel P. Miggins[4] & Daniel E. Heaton [4]

The Easter mantle plume has produced one of the longest hotspot tracks in the Pacific Ocean. While previous studies have focused on the eastern side extending across the Nazca Plate, we use $^{40}Ar/^{39}Ar$ isotopic and geochemical data to investigate the less explored western side around the Easter Microplate. We propose a dynamic model in which a deeper (600 km-depth), less buoyant mantle exerts a westward force on the East Pacific Rise (EPR), while a more buoyant plume region drives Easter hotspot volcanism and a localised acceleration in seafloor spreading. Our findings suggest that the Easter hotspot is the more focused surface expression of the most buoyant region of a vast, deep-seated mantle plume extending from the Pacific Large Low Shear Velocity Province (LLSVP). This challenges the traditional view of hotspots as isolated phenomena and suggests they are part of broader LLSVP-related mantle structures. Our results imply a more intricate, large-scale relationship between hotspots, mantle plumes, spreading ridges, and mantle dynamics.

The numerous volcanic ridges and seamounts scattered across the Pacific seafloor offer valuable insights into the interactions between hotspots and the lithosphere-asthenosphere system (e.g.[1–3]). Furthermore, volcanism associated with near-ridge hotspots provides a useful framework for examining the dynamics between mantle plumes and spreading ridges.

Notably, the Easter Island (aka Rapa Nui) region, situated east of the Easter Microplate on the Nazca Plate and adjacent to the East Pacific Rise (EPR) (Figs. 1, 2), is linked to a deep, lower mantle plume origin (e.g.[1,4–7]). Easter stands out as a rare high buoyancy, high helium ($^3He/^4He$), PREvalent MAntle (PREMA) type plume that closely adheres to the classic plume model, similar to counterparts such as Louisville, Kerguelen, Reunion, and Tristan[1,8]. The hotspot track generated by the Easter plume spans thousands of kilometres across the Nazca and Pacific plates, comparable in scale only to the Louisville Seamount Chain in the southeast Pacific.

The Easter hotspot generated the Salas y Gómez Ridge (SYGR), which extends eastward across the Nazca Plate (Fig. 1). To the west of

the Easter hotspot, the Easter microplate evolved in two tectonic stages[9–11]. From ~5 Ma and until about 2.5 Ma the East Rift propagated northward across the Easter-Salas hotspot, transferring and rotating lithosphere of the Nazca Plate clockwise. The second stage, beginning around 2.5 Ma, involved a slowdown in the northward propagation and continued curved opening of the East Rift, a southeastward propagation of the SW Rift, and the rigid rotation of transferred Nazca lithosphere[9,10].

Published $^{40}Ar/^{39}Ar$ dates for basaltic rock samples dredged from the SYGR and along the Nazca Ridge are consistent with age-progressive volcanism currently originating near Salas y Gómez Island (~106 °W)[12,13]. This agrees with the observed higher Pb and Sr and lower Nd isotopic ratios (i.e., 'enrichment') in basalts from Salas y Gómez Island and adjacent seamounts, compared to lavas from volcanoes further west, including Easter Island, and the East Rift of the Easter microplate[14–16]. The decreasing Sr-Nd-Pb 'enrichment' closer to the East Rift is attributed to the deflection of plume mantle toward the East Rift and progressive melt extraction of enriched mantle

[1]GeoZentrum Nordbayern, Friedrich-Alexander-Universität Erlangen-Nürnberg (FAU), Schlossgarten 5, Erlangen, Germany. [2]Faculty of Science, Vrije Universiteit Amsterdam, De Boelelaan 1085, Amsterdam, Netherlands. [3]Geo-Ocean, UMR6538 Univ Brest / CNRS / Ifremer / UBS, Institut Universitaire Européen de la Mer, rue Dumont Durville, Plouzané, France. [4]College of Earth, Ocean, and Atmospheric Sciences, Oregon State University, Corvallis, OR, USA. ✉e-mail: j.m.oconnor@vu.nl

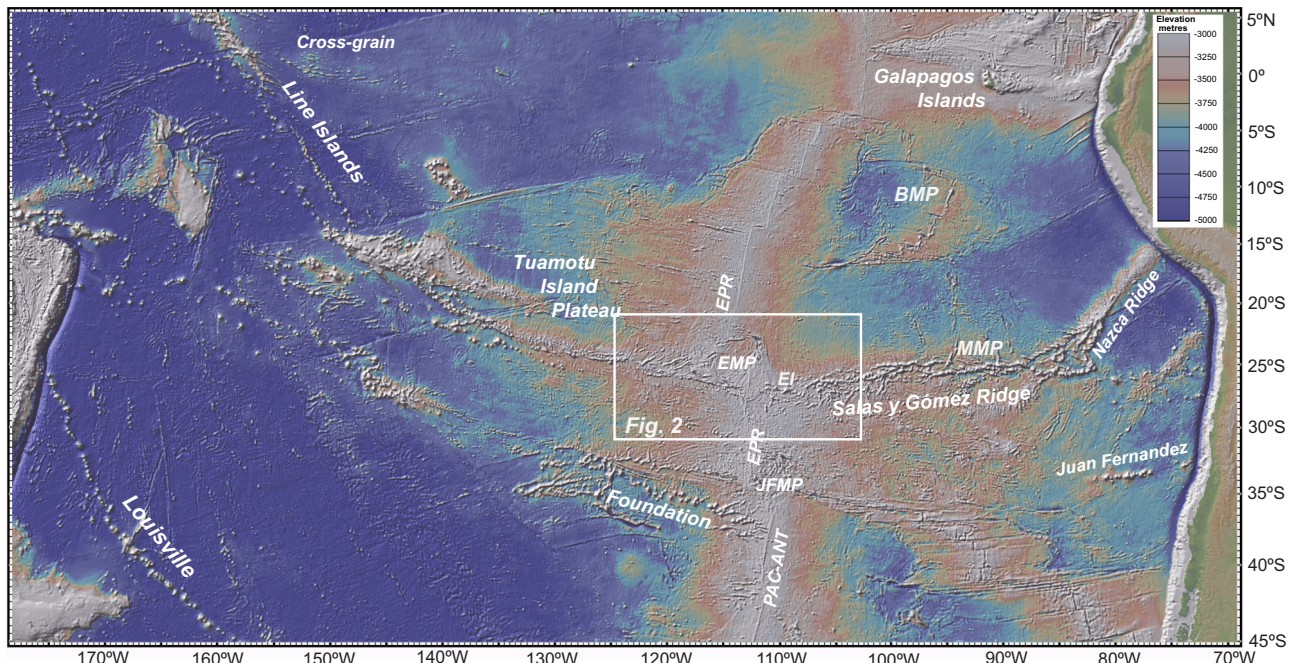

**Fig. 1 | A Pacific-wide view centred on the Easter microplate and the Easter Seamount Chain.** EI: Easter Island; SYG: Salas y Gómez Island; EPR: East Pacific Rise; EMP: Easter microplate; JFMP: Juan Fernandez microplate; MMP: Mendoza Microplate; BMP: Bauer Microplate. Map generated using the default base map in GeomapApp (www.geomapapp.org).

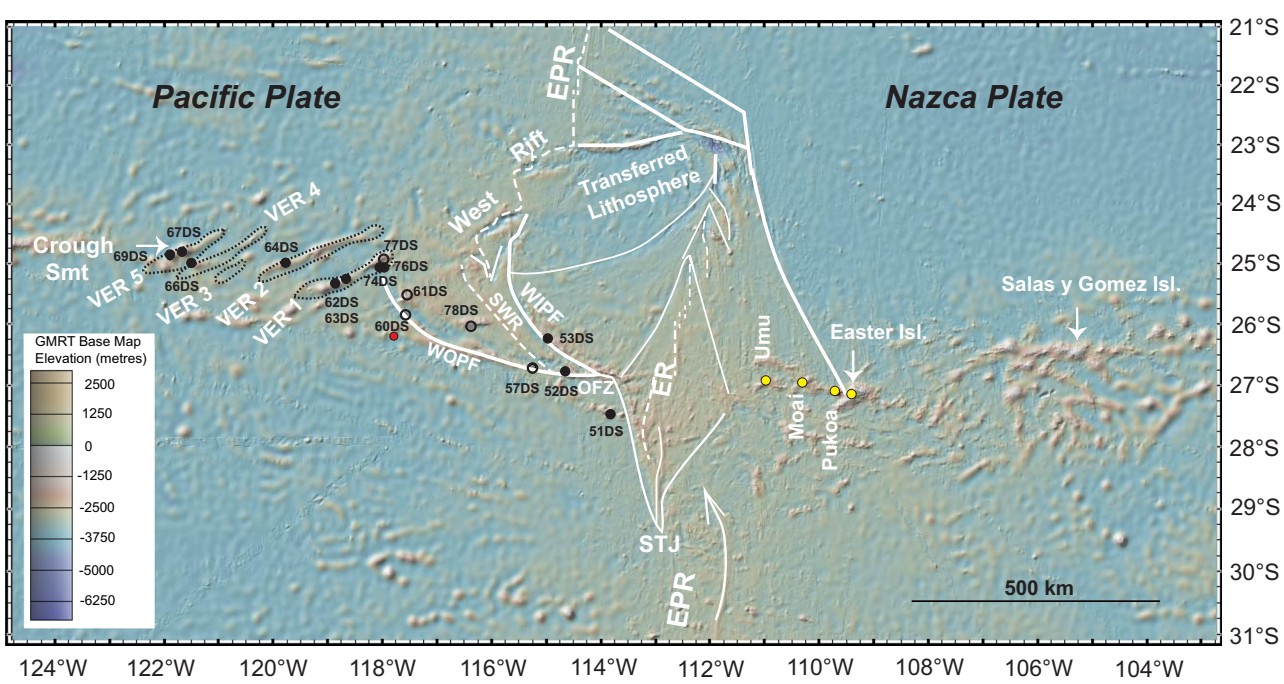

**Fig. 2 | Sample site locations and tectonic features.** $^{40}$Ar/$^{39}$Ar dates and geochemical data are reported for sample sites marked by solid black dots. Geochemical data are available for sites marked with black open circles. Yellow dots are for dredge sites with Ar/Ar dates in the literature[12]. White lines represent a simplified tectonic interpretation showing the main features associated with the Easter microplate - dashed for active spreading ridges[9]. The other tectonic features are after Wilder[44]. ER: East Rift; SWR: Southwest Rift; WOPF: west outer pseudo fault; WIPF: west inner pseudo fault; OFZ: Orango FZ; STJ: Southern Triple Junction; VER: volcanic elongate ridge; EPR: East Pacific Rise. Map prepared in GeoMapApp.

components[14–18]. Alternatively, it may result from plume-ridge mixing and the effects of decreasing lithosphere thickness[19–22].

Numerous multidisciplinary studies have investigated the Easter hotspot track and its association with the EPR. While these studies have generated considerable findings, developing cohesive models that integrate surface observations with deep mantle processes remains a significant challenge. Previous research predominantly focused on the Nazca side of the EPR. Our study bridges this gap by exploring volcanic ridges and seamounts extending from the Pacific plate across the southwest margin of the Easter microplate (Fig. 2). Our primary goal is

to provide new isotopic dates and geochemical analyses for this complex network, aiming to clarify the influence of the Easter plume on microplate development, the SYGR, and the asymmetric spreading history of the slowly migrating EPR (as expressed by the uniquely low ratio of lateral ridge migration relative to the very fast total spreading of the EPR)[23,24]. By synthesising findings from various studies, we aim to construct a model that enhances our understanding of the interactions between plumes from the Large Low-Shear-Velocity Province (LLSVP), surface intraplate volcanism, and spreading ridge migration

## Results

During the RV SONNE 80a-Midplate III expedition (Valparaiso - Easter Island[25]), basaltic rocks were dredged from seamounts and volcanic elongate ridges (VERs) located on the Pacific Plate and the south-western side of the Easter Microplate (Fig. 2)[26,27]. The VERs range from 70 to 200 km in length, in width from 20 to 35 km, and have elevations ranging from 1100 to 2900 meters. Their morphological character-istics resemble the 'cross-grain' type ridges observed in other Pacific regions, which have been associated with plate-wide tensile stress[3,26,28,29]. In this study, we present primarily new isotopic dates and geochemical analyses from these VERs (black dots in Fig. 2) and combine these with existing data (open circles).

Predominantly trending in an ENE direction, the VERs display a right-stepping *en echelon* pattern with an E-W regional trend, resulting in an ~400-kilometre offset of local magnetic anomalies[9,10,26–28,30]. This study focuses on four of the previously described VERs. The south-western ends of the VERs are marked by seamounts, notably Crough and Thomas, located at the ends of VER 5 and VER 4, respectively[26]. The northeastern ends intersect with the lithosphere formed at the SW Rift of the microplate (Fig. 2). Further details on these volcanic edifices can be found in refs. 25,26,28.

### Isotopic Dating

Analytical methods and a discussion of the data quality are detailed in the "methods" section. Summary and complete data files can be found in Supplementary Data 1 and Supplementary Data 2, respectively. Supplementary Data 3 provides a summary the $^{40}Ar/^{39}Ar$ incremental heating isotopic dates. The calculations for $^{40}Ar/^{39}Ar$ dates are presented in Supplementary Data 4. The age-distance trends defined by new dates for the southwest margin of the Easter microplate and published isotopic dates for the SYGR are shown in Fig. 3. Notably, samples from volcanic edifices on the Pacific Plate and the microplate lithosphere (SW Rift) exhibit contrasting age progressions. The 'Pacific trend' (red symbols in Fig. 3) is characterized by dates obtained from the southern ends of the VERs. This trend corresponds to a propagation rate of ~65 mm/yr. In contrast, the 'microplate trend' (blue symbols in Fig. 3) is constrained by dates for samples from the northern end of VER 1, inner and outer pseudofaults of the SW Rift, and from a small ridge or large seamount associated with the ambiguously defined triple junction region in the southern part of the East Rift[9]. This trend indicates a propagation rate of ~192 mm/year, likely reflecting the fast propagation of the SW Rift, and a start of volcanic activity on this ridge, as evidenced by samples from VER 1 prior to 3.5 Ma and continuing at its southern end until about 2.7 Ma.

### Trace Element and Isotope Composition

Analytical methods are in the "methods" section. Trace element, and Sr-Nd-Pb isotopic ratios are provided in Supplementary Data 5. Lavas from the 'Pacific' trend exhibits more 'enriched' trace element com-positions, indicated by higher Nb/Zr and La/Sm ratios compared to NMORB (Fig. 4), and higher $^{87}Sr/^{86}Sr$ ratios (Fig. 5). In contrast, the 'Microplate' trend lavas are less enriched, showing lower Nb/Zr and La/Sm ratios compared to normal MORB (NMORB) (Fig. 4). Furthermore, trace element ratios span a spectrum from depleted MORB (D-MORB) to enriched MORB (E-MORB) compositions. However, the Pb isotope

ratios do not show a marked difference between the 'Pacific' and 'Microplate' trends (Fig. 5), except that samples from Crough Sea-mount (DS67, DS68), located at the southern end of VER 5, display the highest $^{206}Pb/^{204}Pb$ ratios.

A comparison of $^{40}Ar/^{39}Ar$ isotopic dates with seafloor anomaly ages (Fig. 6) indicates that the 'Pacific' trend lavas erupted ~170 km west of the nearest spreading axis, with the underlying crust being around 2 Ma at the time of eruption. Conversely, at the northeastern end of VER 1, associated with the 'microplate' trend, lavas erupted near the closest spreading axis, i.e., the eventual microplate SW Rift. The relatively depleted composition of these 'microplate' lavas correlates with their formation closer to or at the SW Rift of the microplate. As such, the microplate trend reflects reduced pressure and increased melting beneath thinner lithosphere, resulting in lower La/Sm and Nb/Zr ratios. In contrast, beneath the somewhat older and thicker Pacific lithosphere, the average extent of melting is lower, which may have induced preferential melting of more fertile, enriched mantle litholo-gies (e.g.[31,32]) that is resulting in higher La/Sm and Nb/Zr ratios. This distinction is particularly evident when comparing the enriched and depleted characteristics of the south and north ends of VER 1, respectively.

Lavas from the SYGR exhibit a distinct and tightly correlated dataset within the Pb-isotope space. Additionally, they demonstrate a linear Pb-Sr isotope relationship, consistent with a two-component mixing model[16]. These Salas-type lavas display the highest radiogenic Pb isotope compositions, evidenced by $^{206}Pb/^{204}Pb$ ratios exceeding 19.6 (Fig. 5). The Easter-type composition ($^{206}Pb/^{204}Pb > 19.0$) extends from Easter Island westward to the microplate, where it overlaps in composition with MORB erupted at the East Rift at the same latitude as Easter Island. This alignment is also consistent with a spike in $^{206}Pb/^{204}Pb$ and $^{3}He/^{4}He$ ratios (RA > 11) observed in the East Rift from 28°S to 26°S ([33]; Fig. 5B).

This indicates that the 'Pacific' and 'Microplate' age-distance trends also have distinct melting processes, beneath the thicker lithosphere of the Pacific Plate and the thinner lithosphere of the SW Rift, as is evident in the data from VER 1. While Sr-Nd-Pb isotope ratios show minimal variation, they reveal a clear distinction between the more depleted hotspot lavas on the western (Pacific) side of the microplate/EPR and the more enriched lavas on the Nazca side, con-sistent with the well-documented two-component mixing process between the Easter plume and the microplate's East Rift (e.g. [19]).

## Discussion

The 'Pacific' and 'Microplate' trends, identified through our dating and chemical analyses, shed light on the temporal and spatial development of the West Rift in the EPR and the SW Rift of the Easter Microplate. This raises a pivotal question: are these trends causally linked to the asymmetric accretion and migration of the EPR, and the positioning of the Easter plume on the Nazca Plate and its characteristics (buoyancy, temperature and composition)? In the following section, we will explore this connection by examining seafloor ages across the Pacific and Nazca Plates, as well as the Easter Microplate

### Plume−EPR−microplate interconnection

By analysing age-distance relationships for the SYGR ( ~104 mm/yr) and seafloor magnetic anomaly ages, we have reconstructed the spa-tiotemporal dynamics between the EPR and the Easter hotspot, cur-rently positioned at ~106°W, and the EPR (Fig. 6 & S1). Nazca Plate seafloor ages south of the SYGR and the Easter microplate yield a plate velocity of ~101 mm/yr (Fig. 6). Extending this interpretation back to Chron C8n.1 (25.951 Ma) yields a similar plate velocity of 99.5 mm/yr, demonstrating that this relationship has persisted since at least 26 Ma (Fig. S1). The correspondence between the age-distance relationships for the SYGR and EPR south of the SYGR implies that the EPR has consistently been offset from the Easter hotspot since at least 20 Ma,

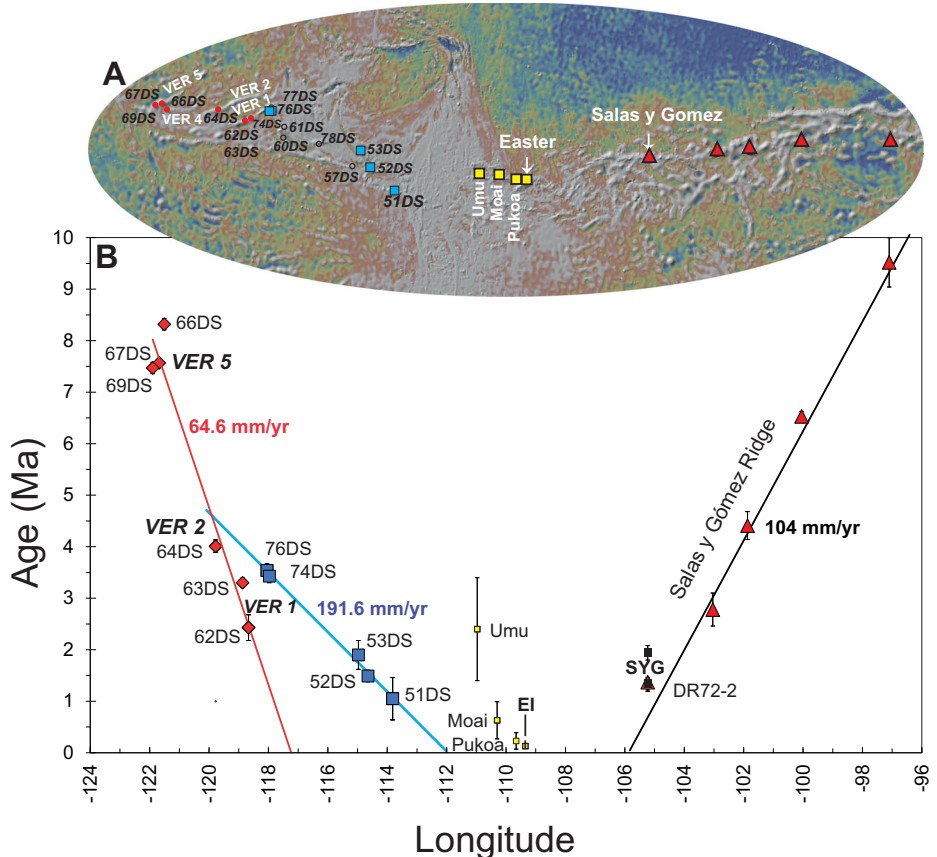

**Fig. 3 | ⁴⁰Ar/³⁹Ar isotopic dates versus longitude. A** Map produced in GeoMapApp showing the VER and microplate morphology. Other details as in Figs. 1, 2. **B** Age-distance relationship for the SO80 sample dates in Supplementary Data 1 defining two diverging trends corresponding to their location on the Pacific (red diamonds) or microplate (blue boxes) crust. The Pacific trend (64.6 mm/yr; solid red line and diamonds) is defined by dates from the southern ends of VER 5 (Crough Seamount; 67DS & 69DS), VER 4 (Thomas Seamount; 66DS), and VER 1 (63DS & 62DS). The date for 64DS at the southern end of VER 2 is aligned with the trend and is not a plateau age (see discussion in the results section). The microplate trend (191.6 mm/yr; diagonal solid blue line and boxes) is defined by dates (blue squares) from the northern end of VER 1 (76DS & 74DS), the inner (53DS) and outer (52DS) pseudo-faults of the West Rift, and a small VER (51DS) associated with the poorly defined triple junction region[9]. VER 1 was erupting and connecting both trends at about

3.5 Ma and continued at the southern end until 2.7 Ma. The microplate trend is interpreted here as a proxy for the propagation of the SW Rift (ref. 9). The red triangles are for published Ar/Ar dates < 10 Ma for samples from the SYGR[13]. Note that sample DR72-2 from the submarine slope of Salas y Gomez is a highly alkalic sample (basaltic trachyandesite). The small yellow boxes represent published ⁴⁰Ar/³⁹Ar dates for submarine volcanism at Umu, Moai, and Easter Island[12]. Additionally, a compilation of published K-Ar ages indicates that the three shield volcanoes on Easter Island−Poike, Rano Kau, and Terevaka−underwent two phases of activity between 0.78 and 0.3 Ma and 0.24 and 0.11 Ma[104]. These phases are marked by vertical black lines. Three K-Ar ages ranging from 3 to 1.89 Ma are considered unreliable, as discussed in Vezzoli & Acocella[104]. The small black symbols are for K-Ar dates (hawaiite & mugearite) for Salas y Gomez Island[105]. Error bars are 2σ.

and very likely since at least 26 Ma (Fig. 6, red bands). This migration rate correlates with ~6.4% more of the crust generated at the EPR accreting to the Nazca side, and an EPR westward migration rate of ~93.6 mm/yr ($V_{EPR} = V_{Nazca} + V_{Pac}$ /2) (e.g.[34,35]) (Fig. S1).

Nazca Plate seafloor ages located north of the microplate between 19°–16° S also yield a consistent velocity (99.7 mm/yr) to the SYGR since at least 26 Ma (Fig. S1). This implies that this segment of the EPR has also been consistently offset from the Easter hotspot since at least 26 Ma. This offset coincides with ~10.9% more crust accreting to the Nazca Plate and indicates a westward EPR migration rate of 89.9 mm/yr (Fig. S1). The constant offset between the Easter hotspot and the EPR, both south of the microplate and to the north between 19° and 16° S, suggests the synchronous westward migration of both the EPR and the Easter hotspot (relative to the eastward migrating Nazca Plate) since at least 26 Ma at a rate averaging ~92 mm/yr (Fig. 6 & S1).

While the EPR exhibits a consistent westward migration rate of ~92 mm/yr, a localised anomaly emerges at the latitude of the Easter microplate. Between the SYGR and roughly 22.5°S seafloor age data suggests a faster age-distance relationship (~112 mm/yr) compared to the regional SYGR rate of 104 mm/yr (Fig. 6, vertical black lines). This

implies that, in this region, the EPR has migrated away from the Easter hotspot at a rate of ~8 mm/yr (112 mm/yr − 104 mm/yr) since at least 13 Ma. This localised anomaly also manifests as an asymmetry in crustal accretion. While the regional average accretion to the Nazca Plate is about 8.6%, in the vicinity of the Easter microplate, it increases to 21.04% on the Nazca side, resulting in a net increase of ~12.5% compared to the regional average. However, it's important to note that despite this local variation, the overall EPR migration rate remains consistent at around 92.7 mm/yr, aligning with the broader regional trend. This contrast highlights a key point: EPR migration appears to be a large-scale phenomenon, evidenced by its consistent average rate. In contrast, asymmetric crustal accretion seems to be a more localised process, potentially influenced by the proximity of the Easter plume to the east (e.g. [35]). We have also used magnetic anomaly data for the Easter microplate (Fig. 7) to establish a correlation between the region-wide synchronous westward (~92 mm/yr) movement of the EPR and the Easter plume and the evolution of the Easter microplate (Fig. 7). The age progression of the 'microplate' lavas (Fig. 3) supports eastward propagation of the SW Rift (Fig. 6A). Eastward propagation of the SW Rift (~191 mm/year; Fig. 3) led to its convergence with the clockwise-

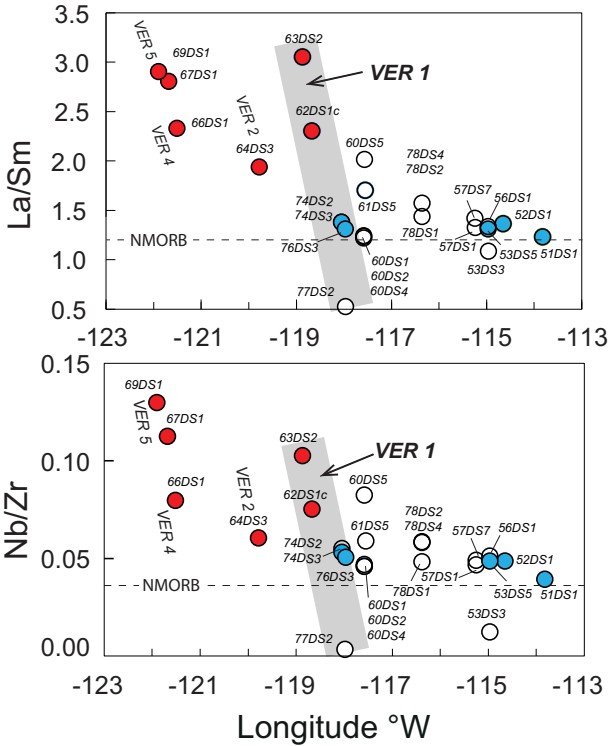

**Fig. 4 | Variation in trace element ratios with longitude.** La/Sm, and Nb/Zr ratios are generally lower for samples that erupted on or closer to the microplate (blue symbols for dated and open symbols for undated) compared to those that erupted on the Pacific plate (red symbols). Dashed lines show average normal MORB (NMORB)[106].

rotating southern tip of the East Rift (Fig. 7) around 0.8 Ma. Subsequently, the East Rift has continued its westward migration (see also Fig. 8). The dashed and solid vertical red lines in Fig. 7 illustrate the correlation between synchronous westward migration of the hotspot (Fig. 7, large red circle) and the EPR and the westward migration/clockwise rotation of the East Rift (Fig. 7, diagonal dashed blue line). Additionally, the figure shows the concurrent anticlockwise propagation of the SW Rift ("Microplate" trend). The relationship between the tectonic evolution of the Easter microplate and the shifting westward position of the Easter plume (and the EPR) since about 5 Ma is further illustrated in Fig. 8. This relationship shows the Easter hotspot/plume traversing across the Easter microplate, driving the rotation and migration of the East Rift and the anticlockwise propagation of the SW Rift.

A role of the Easter plume in the local increase in EPR asymmetric accretion and migration rate relative to the hotspot (~8 mm/yr) aligns with eruption of the 'Pacific' trend lavas on a relatively young seafloor (~2 Ma), located about 170 km from the West Rift (EPR) (Fig. 6). Notably, the oldest 'Pacific' trend lavas suggest an Easter-like composition (Fig. 5, Crough Seamount) prior to the formation of the microplate. Isotopic dating also indicates that volcanic activity along the 'Pacific' trend might have ceased around 2.5 Ma, shortly after the initiation of the SW Rift at about 3.5 Ma (Fig. 6). This effectively isolated the Pacific Plate lithosphere from the influence of the Easter hotspot. This pattern suggests that before the curved opening and eastward propagation of the SW Rift around 3.5–4.0 Ma (Fig. 3)[9,10], the Easter hotspot significantly influenced the formation of seamounts and ridges. Moreover, lavas from the microplate show relatively depleted incompatible element and isotope compositions, aligning with those from young lavas on the West Rift axis (Fig. 5A).

In summary, regionally, the EPR maintains a constant distance from the westward migrating Easter plume, implying that they are both

moving westward at the same speed of ~92 mm/yr (relative to the eastward drift of the Nazca Plate). This results in an ~8.9% asymmetrical accretion to the Nazca Plate. However, at the latitude of the Easter microplate, a ~12.4% increase in EPR asymmetry coincides with the migration of the EPR (current West Rift) away from the hotspot and the formation of the Easter microplate starting around 5 Ma. This excess accreted crust, including the Easter Microplate, is in the process of being significantly overprinted by the more buoyant upper mantle expression of the Easter plume as shown seismically (see below) as it is transferred to the Nazca side of the EPR and the SYGR. The relationship between the tectonic evolution of the Easter microplate and the westward shift of the buoyant upper mantle hotspot since about 5 Ma (Fig. 8) illustrates this process in action.

**Two-stage development of the SYGR**

Salas-type lavas, extending along the SYGR as far west as Easter Island (Fig. 5), mark the current centre of the 700 km diameter surface hotspot (see below). These lavas align with deeper, more enriched melting processes above older, thicker seafloor lithosphere, indicative of late-stage volcanic activity. They predominantly emerge on the eastern ('age progressive') side of the surface hotspot, presumably sampled from the upper flanks of the Salas y Gómez Ridge (SYGR). This origin contributes to the distinct linear age progression observed along this ridge (Fig. S1). In contrast, Easter-type lavas, found to the west, exhibit a chemical gradient and younger ages. This pattern is consistent with shallower and more extensive melting processes, facilitated by the decreasing age of the underlying lithosphere closer to the East Rift. These lavas emerge across the plume on young, thin oceanic lithosphere near the ridge in a non-age progressive manner. Their compositions result from the deflection of the plume mantle, progressive extraction of enriched mantle components, and the thinning effects on the lithosphere.

Various general observations influence our understanding of the role of mantle plumes and plate motion in the formation of micro-plates. In particular, the EPR accommodates its offsets via microplates, propagating rifts, or overlapping spreading centres, with a noted absence of transform faults beyond certain spreading rate thresholds (ref. 9). Moreover, the interaction of hotspots with the EPR creates regions of hot and relatively thin lithosphere facilitating rift propagation and microplate formation due to enhanced stress changes (refs. 36,37). Furthermore, the formation of the Easter and Juan Fernandez microplates, which began around 5 Ma, coincided with a significant reorganisation of tectonic plates in the circum-Pacific region (e.g.[11,38,39]). This timing suggests that some microplates may be initiated by such plate reorganisations or short-lived, abrupt northward shifts in the direction of the Pacific Plate[29].

South of the SYGR, at least nine paleo-microplates have been conclusively identified, all of which have undergone a transfer process to the Nazca Plate, with the most recent and actively evolving being the Juan Fernandez microplate (ref. 40, *their Figure 16.6*). These paleo-microplates are primarily identified through satellite data, which reveals distinct curved seafloor patterns attributed to their rotation[41], and the existence of curved compressional ridges, some of which reach elevations of up to 1 km (refs. 10, 40, 42). Our findings suggest these paleo-microplates formed near the EPR south of the SYGR, migrating westward in conjunction with the deeper Easter plume mantle below about 600 km depth. They lack prominent surface expressions of the hotspot unless the plume becomes more buoyant above 600 km. This suggests that paleo-microplates are incorporated into the SYGR and may have undergone volcanic (and geochemical) overprinting, obscuring their distinct microplate identity. To address this issue, we use a series of schematic reconstructions (Fig. 9), exploring the possible history of paleo-microplates and excess crust accretion to the Nazca Plate and the SYGR (Fig. 9). This analysis considers both the vast region of plume mantle below 600 km extending beyond the EPR, as well as the hotter, more buoyant plume mantle

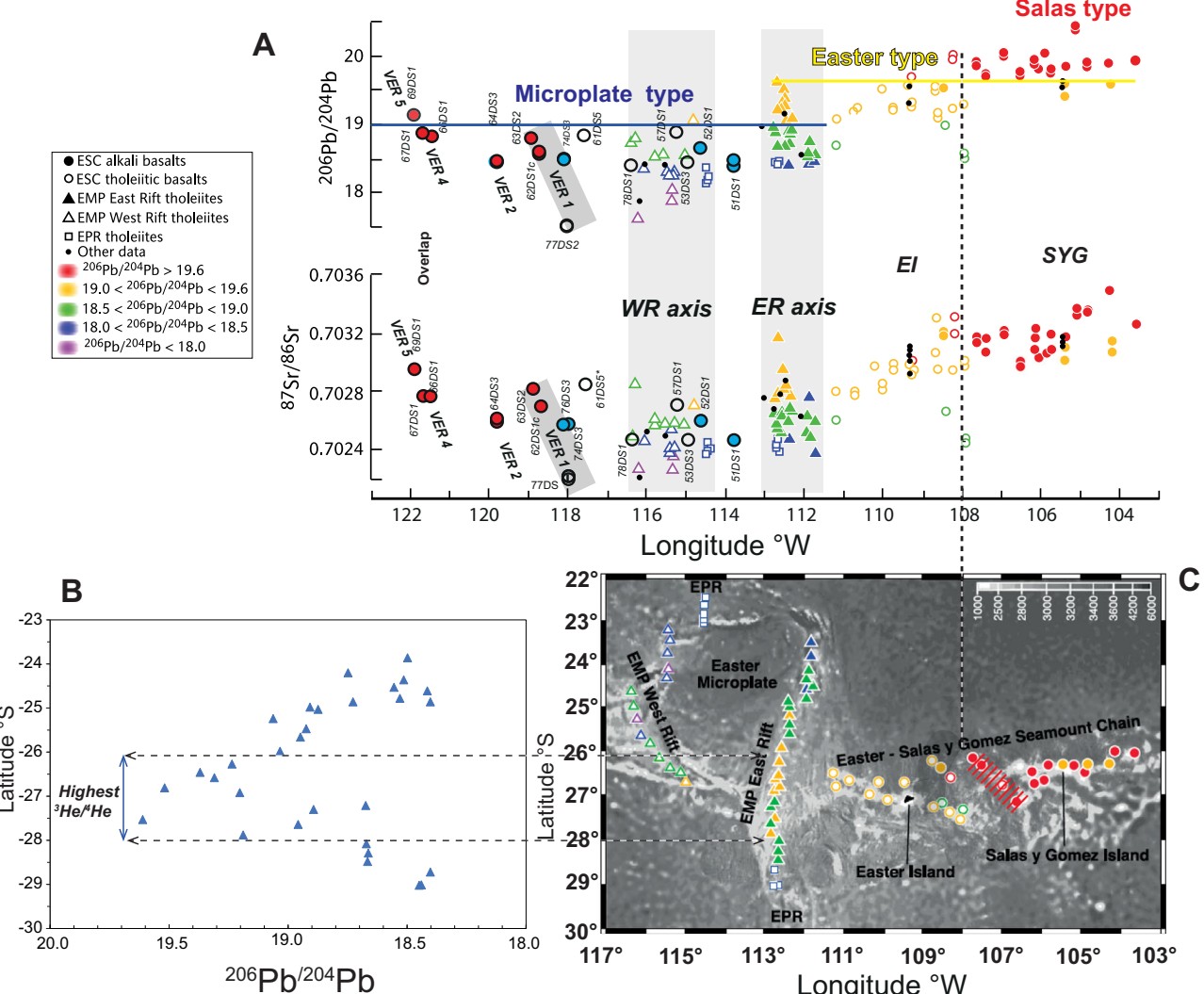

**Fig. 5 | Variation in isotope ratios with longitude. A** $^{86}Sr/^{86}Sr$ and $^{206}Pb/^{204}Pb$ data for basalts from the microplate spreading rifts are from ref. 16. Literature symbols are colour-coded based on $^{206}Pb/^{204}Pb$: Red > 19.6; yellow > 19.0. High values indicate a greater proportion of a hotspot–related component relative to a depleted upper mantle component in the mixtures that produced the basalts. For discussion, we term these levels of enrichment as 'Salas' and 'Easter' types, respectively. The literature and SO80 data < 19.0 are referred to as the 'microplate' type. The vertical dashed black line indicates the transition from Salas to Easter type source ~150 km east of Easter Island at ~108°W (see also light blue dots in Fig. 10). Other details are the same as in Fig. 4. **B** $^{206}Pb/^{204}Pb$ data for the East Rift versus latitude[16]. Horizontal dashed lines denote the peaks in $^3He/^4He$ ratios ( > 10 RA) and $^{206}Pb/^{204}Pb$ between 26° and 28°S in lavas from the East Rift[33]. **C** Sample location map from ref. 16. NW-SE dashed red line shows the broadness of the Salas anomaly.

concentrated on the Nazca Plate side of the EPR between the East Rift and Salas y Gomez Island, above 600 km (Fig. 10).

The Mendoza paleo-microplate stands out as one of the most substantial microplates associated with the SYGR, evolving roughly between 20 Ma and 6 Ma[43,44]. It initiated around 20 Ma with the formation of the Mendoza Rise, which propagating northward[43,44] from the latitude of the SYGR. Between ~16 Ma and 11 Ma, the southern ends of the Mendoza Rise and evolving Mendoza microplate were traversed by the surface hotspot (Fig. 9, black dot). This relative motion is evident from the correlation around 11 Ma between the hotspot's eastern edge and the youngest seafloor anomaly age associated with the Mendoza Rise (Fig. S1). Significantly, the EPR was migrating away from the hotspot during this same period (Fig. 9, blue dots), generating excess crust that potentially evolved into a microplate. By about 11 Ma, the southern end of the Mendoza microplate reached the eastern side of the surface hotspot, while the western side of the hotspot traversed the region of excess crust/microplate as the EPR continues migrating away ( ~ 8 mm/ yr; Fig. 6). Around 8 Ma, this process led to the onset of the 'Pacific

'trend and the VERs and by around 5 Ma to the initiation of Easter microplate (propagation of the East Rift). This model aligns with the previously discussed relationship between VERs, and waning plume-enhanced plate tensile stress associated with rift propagation and microplate formation. As the EPR continued migrating westward from the plume after 8 Ma, further interactions with a propagator from the south around 5 Ma (East Rift) and the EPR to the west (West Rift/SW Rift) culminated in the formation of the Easter microplate. The inception of VER 1 starting around 3.5 Ma reflects the complex dynamics between these segments of the EPR. In contrast, the Galapagos Rise maintained a stable relationship with the northern EPR segment from about 20 Ma until the transfer of the Bauer microplate to the Nazca Plate around 6 Ma[45] (Fig. 10, large white dots). Notably, the EPR does not extend beyond the western limit of the deeper plume upwelling region below 600 km extending beyond the EPR (Fig. 9, dashed yellow lines).

In summary, the transition from Easter to Salas composition observed in the SYGR, along with the schematic reconstructions support a two-stage accretion model spanning at least 20 Ma. The first

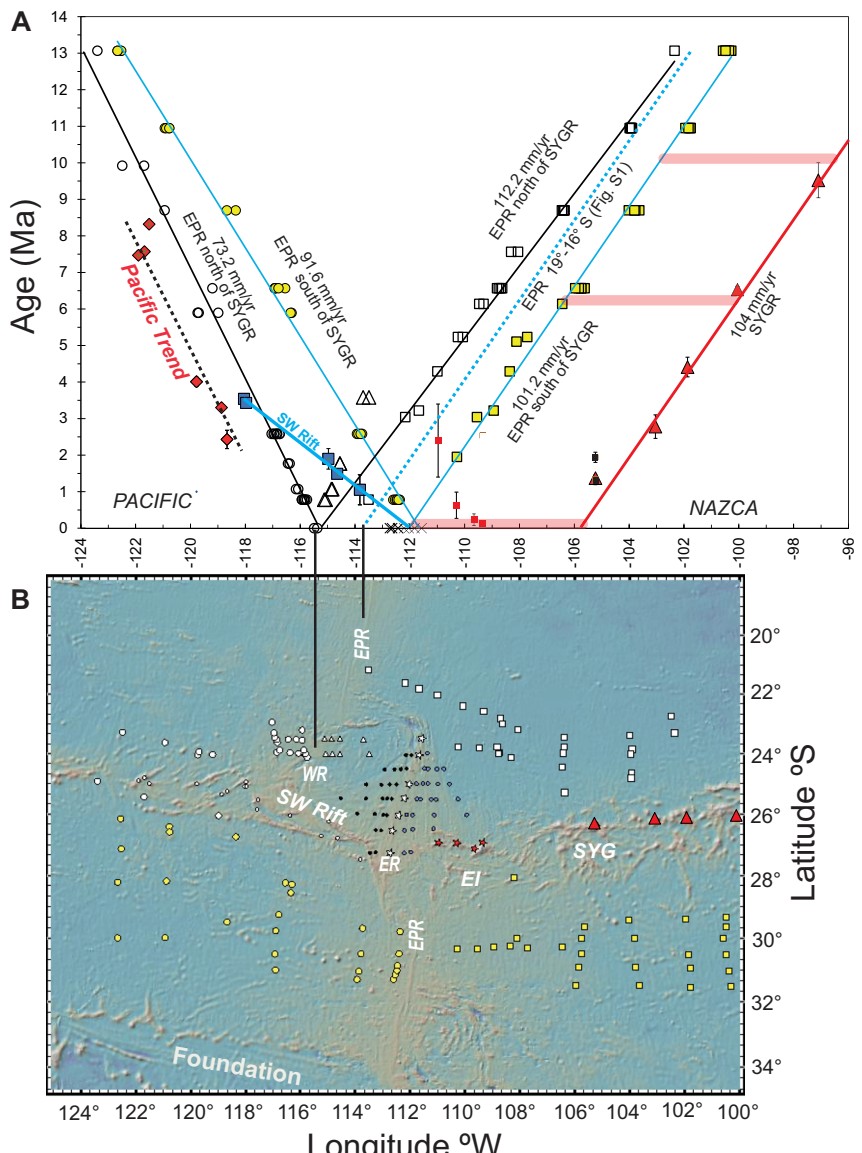

**Fig. 6 | Relative motion between the East Pacific Rise and the Easter hotspot. A** Magnetic anomaly ages versus longitude using the time scale in GEEK2007[107]. The blue and black trendlines are fitted to Nazca anomaly picks flanking the SYGR to the south and north, respectively. The similar length of the red bands signifies a constant offset between the hotspot (Fig. 3 & S1) and the axis south of the SYGR. The dashed version of this blue trendline and the short vertical black line show that north of the microplate the current position of the EPR agrees with consistent EPR migration relative to the hotspot. The assigned numbers correspond to the half-spreading rates in mm/yr. The diagonal solid blue line represents the inferred propagation of the SW Rift based on SO80 Ar/Ar dates discussed in the text (Fig. 3). The ⁴⁰Ar/³⁹Ar dates defining the 'Pacific' trend (red diamonds and black dashed line) align with the migration of the EPR (West Rift) crust (circle symbols and black

regression line). The distance between these trends implies that the 'Pacific' trend volcanism erupted about 170 km west of the EPR (West Rift). Asymmetric crustal accretion to the Nazca Plate at the latitude of the hotspot amounts to ~21%, while to the south, it is ~6% (Asymmetry Percentage = [(difference in spreading rates / total spreading) * 100]). The long vertical back line denotes the resulting current location of EPR at the latitude of the hotspot. The ⁴⁰Ar/³⁹Ar ages defining the SYGR age progression[13] are indicated by the red triangles (see also Fig. S1). **B** The map shows the location of magnetic anomaly picks since anomaly C5AAn (13.065 Ma) from ref. 108, http://www.soest.hawaii.edu/PT/GSFML/. The box and circle symbols represent isochron picks on the Nazca and Pacific plates, respectively. Yellow shading indicates picks south of the Easter hotspot system. Symbols for picks on the Easter microplate are explained in Fig. 7.

stage involves the generation of excess crust and microplates west of the surface hotspot, possible triggered by plate motion changes. The second stage involves significant volcanic and geochemical over-printing influenced by the movement of the Easter plume and its buoyant upper mantle expression. This overprinting explains the persistence of the Salas-type compositions along the age-progressive segment of the SYGR. The latest cycle of this two-stage SYGR accretion process began around 8 Ma with the onset of the 'Pacific' trend, followed by the VERs, and culminating in the formation of the Easter microplate around 5 Ma.

## Tilted plume mantle from the Pacific LLSVP

The ~700 km diameter Easter plume, located between the current Salas hotspot (~106°W; Fig. 3 & S1) and the microplate's East Rift ([6]; Fig. 10), features the slowest (~1%) (hottest/most buoyant) seismic anomaly centred at Easter Island (Fig. 10). At depths between 600 km to 1000 km, the plume is located roughly 350 km west of the EPR. As it rises towards the surface, the Easter plume becomes more buoyant, showing an increasing focus towards the eastern side of the East Rift. To the north of the microplate, another region of warm mantle characterised by higher seismic velocity (i.e., cooler/less buoyant) extends

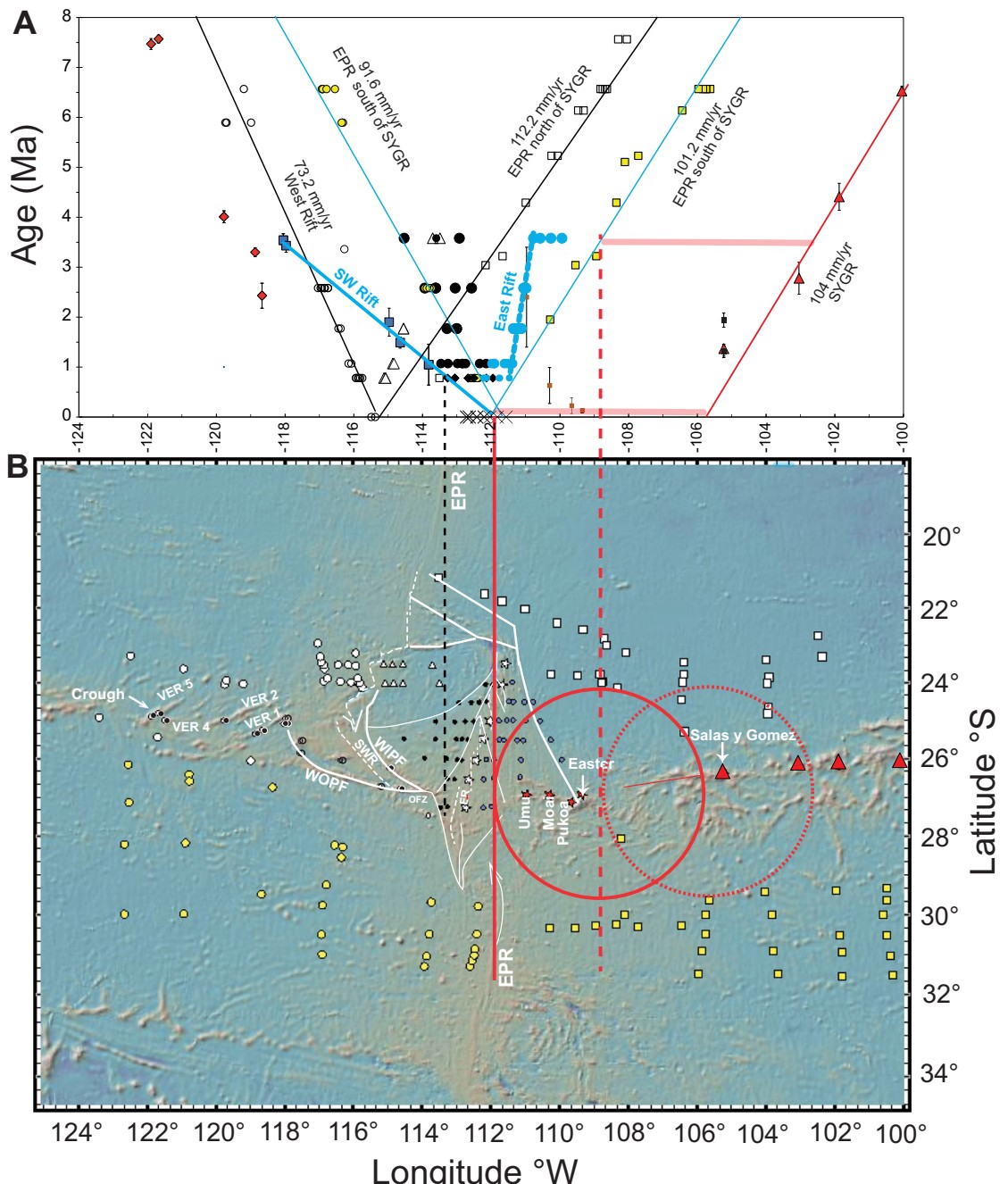

**Fig. 7 | Relative motion between the evolving Easter microplate and the Easter plume. A** Magnetic anomaly ages and isotopic dates are plotted versus longitude. The dashed diagonal blue line is for the propagation and clockwise rotation of the (north end) of the East Rift (see also Fig. 8). The vertical dashed and solid red lines are for the western side of the Easter plume and the EPR to the south at 3.5 Ma and 0 Ma, respectively. Note the convergence of the East Rift with the west side of the plume and the EPR to the south between 3.5 Ma and 0.8 Ma. The vertical dashed black line represents the corresponding opposite propagation of the SW

Rift until around 0.8 Ma. Other details as in Fig. 6. **B** This map is like 6B. Small black and blue dots represent picks on the west and east sides of the East Rift, respectively. The triangles are associated with picks on the transferred and rotated Nazca lithosphere and on the crust formed on the microplate side of the West Rift. The white stars are for 0 Ma picks defining the location of the East Rift. The solid red circle is for the Easter plume at the plate surface (as defined seismically in Fig. 10). The dashed red circle is for the location of the plume at 3.5 Ma.

about 350 km beyond the EPR at similar depths. As it ascends, this material increasingly focuses beneath the central region of this segment of the EPR, though it is no longer detectable above the 300-kilometre in the global DETOX-P2 model (see Fig. 10, Profiles D-F). However, localised geophysical and geological studies on the western flank of the EPR segment between 19°S to 12°S —such as the 'MELT' and 'GLIMPSE' study areas—have revealed phenomena, including subsidence, volcanic activity, significant geochemical diversity and

seismic velocity and density anomalies down to at least 600 km (refs. 46–53). Notably, these geological formations contribute to the creation of swells like the Hawaiian Swell.

The DETOX-P2 model also indicates that EPR is aligned above a deep mantle (2800 km-depth) north-south striking belt of 'slow' mantle extending from the Mexican west coast to ~40°S (ref. 6; Fig. 10 & S2). This region is a part of the 'Pacific large low shear velocity province' (LLSVP), considered to be a plume-generation zone[54,55]. The

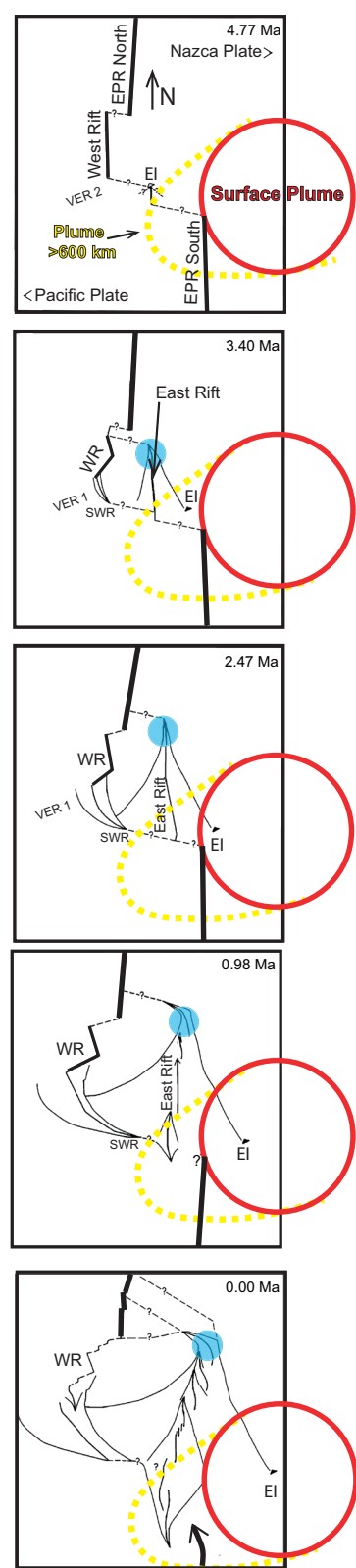

**Fig. 8 | Tectonic reconstruction of the Easter microplate.** Schematic illustration depicting the synchronous westward shift of the Easter plume system and the EPR, alongside the tectonic evolution of the Easter microplate. The panels show the formation and evolution of the Easter microplate from[9]. Red circles represent the buoyant Easter plume at the surface (hotspot), while dashed yellow lines represent the deeper, less buoyant plume extending beneath and beyond the EPR (as in Fig. 10). The surface hotspot and the EPR to the south are shown migrating westward together while the West Rift is migrating away (~8 mm/yr; Fig. 6). The microplate forms via the northward propagation of the East Rift and the corresponding southeast propagation of the SW Rift. Note that as the East Rift propagates northward until about 2.5 Ma, the hotspot is moving closer. After the East Rift stopped propagating northward, the southern region is pushed away and rotated clockwise in sync with the advancing plume. The northern end of the East Rift is now rotating closer (blue dot) to the encroaching hotspot. Using Easter Island as a reference point, it is evident that as plume is traversing beneath the microplate, it is being volcanically and geochemically overprinted by the more buoyant surface hotspot. This illustrates the proposed 2-stage accretion model for the SYGR as discussed in the text. The dynamic relationship between evolving microplate and the plume at different depths is evident from the relationship between the southern extremity of the East Rift and the extension of the plume beyond the EPR below 600 km. The top, bottom, left, and right of the 0 Ma panel are located at 21°S, 30°S, 118°W, and 109°W, respectively. Further details about reconstruction are in ref. 9.

conduits reach deep into the mid-mantle, where they extend outward and upward, ultimately connecting to hotspot locations[56–58].

In summary, mantle tomography data reveal the presence of at least two plumes with a downward tilt towards the southwest. The EPR has maintained a constant offset from both of the plumes while drifting westward at ~92 mm/yr (relative to the Nazca Plate) since at least 26 Ma (Fig. S1), implying that they are moving westward at the same speed. These plumes originate from the same area within the Pacific LLSVP and extend under and beyond the EPR to depths below about 600 km. Of these two plumes, the Easter plume is notably more buoyant within the upper mantle, predominantly focusing on the Nazca side of the EPR. However, due to resolution limitations, particularly in the mid-mantle as those discussed by Hoseini et al. (2020), it remains uncertain whether these represent two distinct plume branches anchored to the LLSVP or are more focused and buoyant regions of a single, vast region of plume upwelling anchored to the Pacific LLSVP. Notably since ~16 Ma, excess crust and microplates have evolved west of the Easter plume, above areas where cooler and weaker plume mantle dips below and extends beyond the EPR (Fig. 10, dashed yellow lines). Additionally, the relationship between the southern end of the Galapagos Rise (Fig. 9, large white dots) and the EPR from about 20 Ma to 6 Ma agrees with the proposed connection between a widespread region of deeper, cooler, weaker plume mantle and large-scale EPR asymmetry. In conclusion, the increased buoyancy and potential temperature of the Easter plume may contribute to the observed maximum spreading asymmetry in the Pacific, as discussed in the next section.

## Variably buoyant plume and asymmetric spreading

Our findings reveal a connection between the synchronous westward migration of the EPR (~92 mm/yr relative to the Nazca Plate) and the Easter plume, along with the asymmetric crustal accretion favoring the Nazca Plate (averaging ~8.9%). This correlates with the presence of a vast plume mantle region extending westward beyond the EPR boundaries, located below about 600 km depth and extending beyond the EPR boundaries (Fig. 10). The observed westward migration of the EPR in the 'MELT' and 'GLIMPSE' study areas on the Pacific side of the EPR aligns with global observations that fast to intermediate spreading ridges tend to migrate toward hotter flanks with more intense magmatic activity[59,60]. This supports the idea that asymmetries in mantle temperature, partial melt concentration, and plume-driven asthenosphere flow contribute to long-term spreading rate asymmetries[23,61].

The significantly more buoyant (hotter/more fertile) Easter hotspot above ~600 km depth coincides with the EPR's migration away

Easter plume ascends south-westwards from a broad low-velocity conduit anchored to the LLSVP, situated roughly midway between the Easter and Galapagos hotspots (Ref. 6; Fig. 10 & S2). Additionally, a weaker and cooler secondary plume also exhibits a downward tilt under the |EPR, originating from the same region within the LLSVP (Fig. 10). This plume system agrees broadly with tomography studies of the African LLSVP, revealing a distinct pattern characterised by one or more low-velocity conduits originating from the LLSVP. These

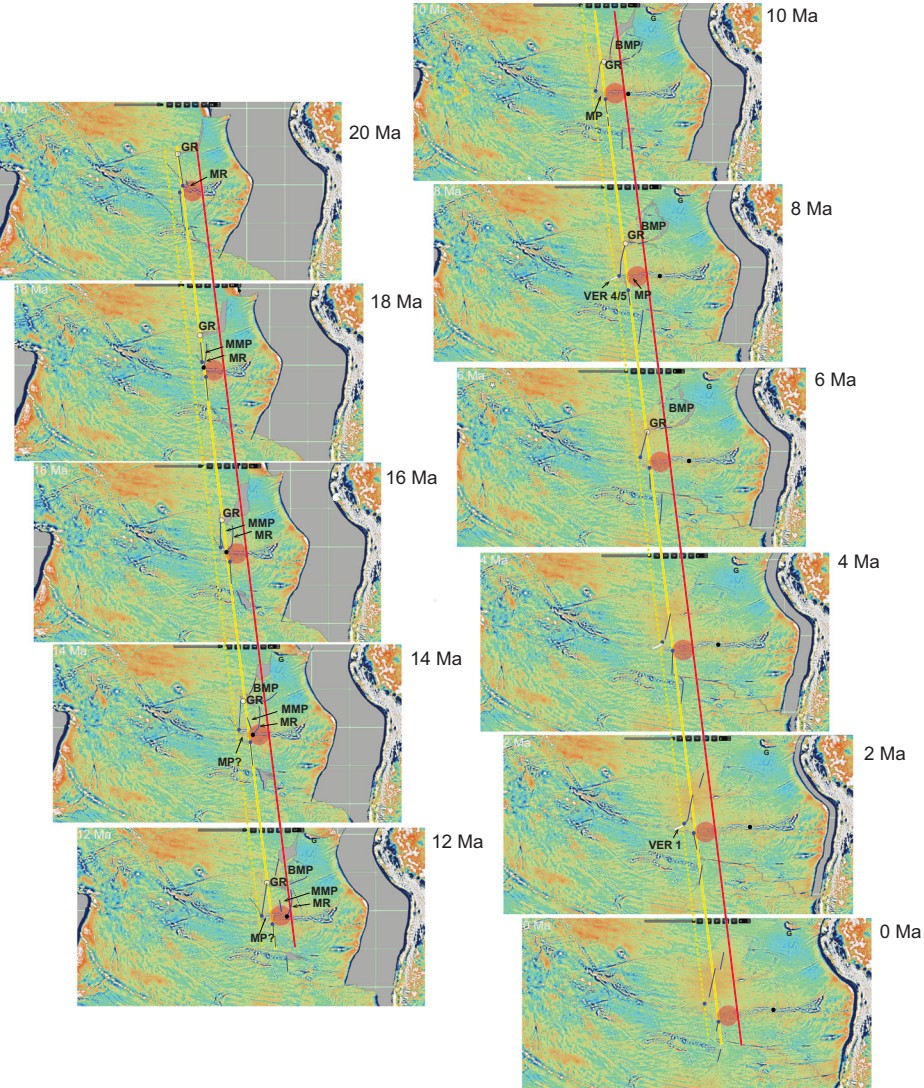

**Fig. 9 | Plate motion and asymmetric evolution of the Salas Y Gomez Ridge.**
Global Free-Air Gravity reconstructions of the relative motion between the Pacific and Nazca plates generated using GPlate software illustrating the complex dynamics of the SYGR accretion process (https://portal.gplates.org/portal/welcome/). The large red discs (with 42% transparency) and solid red and yellow diagonal lines define the eastern and western boundaries, respectively, of the surface Easter mantle plume (Fig. 10). The dashed yellow lines depict the plume's projection ~ 350 km beyond the EPR at depths below about 600 km. Solid blue lines and dashed blue lines indicate, respectively the position of the EPR and pseudofaults that trace ridge propagation. The Nazca-Pacific reconstructions assume that the EPR north and south of the SYGR is coincident with the eastern edge (yellow diagonal lines) of the Easter plume (red disks). Black dots denote the changing location of the propagator that evolved west of the Mendoza Rise. Blue dots mark the EPR segment west and south of the Easter plume. Large white dots show the constant offset between the EPR and the southern end of the Galapagos Rise/Bauer Microplate[46]. MMP: Mendoza Microplate; MR: Mendoza Rise; MP: microplate; BMP: Bauer Microplate; G: Galapagos Islands; GR: Galapagos Rise.

from the hotspot. This results in a local nearly doubling of asymmetric crustal accretion compared to the regional EPR average and facilitating microplate formation. This buoyant upper mantle plume east of the EPR weakens the relatively thin lithosphere, leading to enhanced stress changes and, ultimately, microplate formation (refs. 36,37). Notable, the absence of transform faults beyond certain spreading rate thresholds is consistent with the systematic transfer of excess crust to the SYGR (ref. 9).

Other proposed mechanisms for asymmetric spreading near plumes have been proposed. These include asthenosphere flow leading to excess melt generation and dyke intrusion[62–67]. Variations in Easter plume buoyancy likely play a key role in asymmetric spreading and microplate formation through several mechanisms (e.g.[23]). The buoyant plume material exerts a force on the surrounding mantle, potentially pushing the Nazca Plate westward (plume push) or dragging the base of the overlying Nazca Plate (basal drag), along with thermal effects, and potentially geochemical factors such as mantle fertility and volatile content (e.g.[23]). The tilted orientation of the Easter plume might exert a stronger force on the Nazca Plate side of the EPR compared to the Pacific side, contributing to the observed faster spreading on the Nazca Plate side. This could contribute to the observed faster spreading on the Nazca Plate side. The additional buoyancy above 600 km likely influences the two-stage volcanic and geochemical SYGR accretion. It might also contribute to the significant northward migration and clockwise rotation of the EPR and the Farallon Ridge over the past 50 Ma, as discussed by Wessel, ref. 68 and refs. 23, 69.

## Absolute plate motion

The mismatch between volcanic propagation along the SYGR and Nazca Plate velocities predicted by plate motion models hinders the

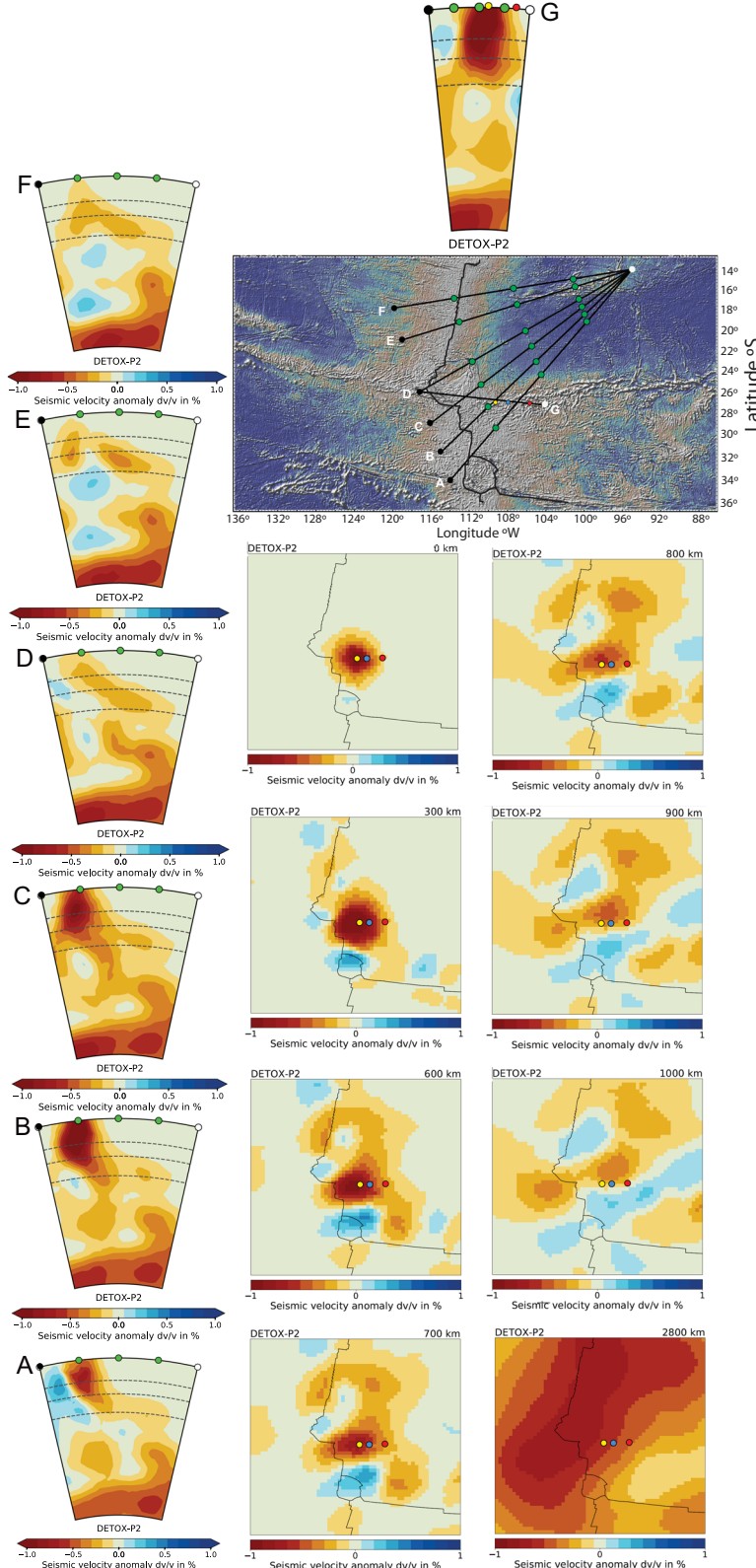

**Fig. 10 | Southeast Pacific LLSVP plume plumbing system.** The vertical column of panels on the left side are DETOX-P2 vertical sections (**A**–**G**) down to the LLSVP (2800 km) (ref. 6). See Figure S2 for a large-scale view encompassing the Galapagos Islands. The topographic map shows the location of profiles (**A**–**G**) relative to the Easter Microplate, the EPR and the SYGR. Panels below are map views at different depths with corner coordinates -40°S; −125°W and −12°S, −95°W. Light blue dots indicate a reference location at 27°S:108°W. Yellow dots mark the location of Easter Island at −26.92°S: 109.35°. Red dots at −27S 105.75 W is the eastern edge of the present plume location defined by SYGR age progression. Figures were generated using 'SubMachine' software[109].

refinement of absolute plate motion (APM) models, crucial for understanding the movement of tectonic plates relative to Earth's deep mantle. The estimated speed of the Nazca Plate implied by the ~104 mm/year age progression along the SYGR (Fig. 3 & S1) is significantly faster that velocity predictions derived from plate motion models. For example, NNN-NUVEL-1A[70] and NNN-MORVEL56[71] predict eastward movements of the Nazca Plate at 80 and 75 mm/yr, respectively. Space geodetic models like GEODVEL10[72] suggest a velocity of 69 mm/yr, while regional GPS measurements confirm a maximum convergence rate of ~66 mm/yr between the Nazca Plate and South America[73]. Fixed hotspot models such as HS2-NUVEL-1A[70,74] and HS3-NUVEL-1A[75] project even slower velocities of 47 mm/yr and 33 mm/yr, respectively. Adding to the complexity, the T25M model, which considers absolute plate motions relative to seismically identified deep mantle plumes (including the SYGR), predicts a velocity of 57 mm/yr for the Nazca Plate[76]. Notably, T25M predicts significantly lower plate velocities compared to those implied by most of the 25 observed age progressions used in its development. This finding agrees with a broader observation: hotspot tracks typically project in the opposite direction of plate motion, converging towards mid-ocean ridges. This observed pattern is consistent with our research reported here. The synchronous westward migration of the EPR and the Easter plume at 92 mm/yr reflects the relative motion between the plume movement and the Nazca Plate's eastward drift. Using Nazca Plate velocity estimates from various models allows us to infer the range of likely estimates of the plume's independent westward motion. For instance, using the Nazca Plate velocity obtained from GPS measurements (~69 mm/yr) or the recent T25M moving hotspot model (~57 mm/yr), combined with the average EPR migration rate (~92 mm/yr), yields plume migration rates of 25 mm/yr and 35 mm/yr, respectively. These inferred plume migration rates align well with mantle flow models[77] that predict a westward motion of the Easter hotspot relative to Hawaii and Louisville at a few centimeters per year.

## Broader Implications of two-stage accretion

The schematic reconstructions in Figs. 8 and 9 depict the complex, two-stage asymmetric magmatic and geochemical accretion process of the SYGR. This process involves initially the formation of excess crust (~12.4%), primarily in the form of microplates. Subsequently, these regions experience significant volcanic and geochemical overprinting, heavily influenced by the Easter plume's movement and buoyancy as they incorporate into the SYGR. These microplates, in turn, significantly influence the distribution and composition of intraplate magmatism across space and time, depending on factors like distance from hotspots and the age/thickness of the lithosphere.

The high buoyancy flux and potential temperature of the Easter plume suggest a link to the underlying Pacific Large Low Shear Velocity Province (LLSVP)[4,78,79]. Lavas associated with this plume exhibit exceptionally elevated $^3He/^4He$ ratios, with values ranging from 17.6 to 26 RA (RA stands for the atmospheric $^3He/^4He$ ratio of $1.39 \times 10^{-6}$)[33,79–81]. High $^3He/^4He$ ratios exceeding 10 RA are observed in lavas from the East Rift between 26° and 28°S, correlating with spikes in both $^{206}Pb/^{204}Pb$ and $^{87}Sr/^{86}Sr$, marking the region where the Easter hotspot influences the spreading ridge ([33]; see Fig. 5).

Further north along the EPR, between 16° and 18°S, lavas associated with the weaker northern plume region also exhibit high $^3He/^4He$ ratios (above 10 RA)[53]. This suggests a connection to the underlying Pacific LLSVP, even though these lavas lack the distinct strontium and lead isotope signatures characteristic of the Easter plume itself[53,82]. Interestingly, Easter-type lavas were found about 400–500 km west of the EPR, specifically from the Hotu-Matua Volcanic Province[51]. These lavas have been dated at around ≤6 Ma using the $^{40}Ar/^{39}Ar$ method[48]. These dates suggest they formed contemporaneously with the Easter and Juan Fernandez microplates. This

implies that stress-related volcanic activity in this older, off-axis seafloor tapped into the composition of the cooler, weaker plume mantle underlying the Easter plume, resulting in these Easter-type geochemical signatures.

The observed correlation between plume buoyancy and variations in helium, along with Sr, Nd, and Pb isotopes, potentially derived from a common LLSVP source, can be attributed to helium's distinct behaviour compared to these other elements[33,53]. During increased melting within the plume, helium is less readily depleted compared to Sr, Nd, and Pb. This is because helium preferentially gets incorporated into melts at higher degrees of melting. Consequently, helium becomes more concentrated within the remaining plume mantle source, leading to the observed distinctive isotopic signature.

Integrating this perspective with the two-stage SYGR accretion model provides insights into the isotopic variations observed along hotspot tracks. It highlights the key role of plume buoyancy, melting depth within the plume, and interactions with the overlying lithosphere in controlling these isotopic variations. The two-stage model developed for the SYGR therefore has implications for understanding the bilateral asymmetry in the geochemistry of other hotspot tracks linked to other Pacific LLSVP-related plumes like Galápagos, Hawaii, Marquesas, Samoa, and the Societies[8,83].

Additionally, the model agrees with the concept of distinct mantle reservoirs linked to the LLSVP, characterised by high $^3He/^4He$ and intermediate Sr, Nd, and Pb isotope ratios known variably as PREMA (PREvalent MAntle)[84], FOZO (Focus Zone)[85], C (Common component)[86] or PHEM (Primitive HElium Mantle)[87]. An alternative perspective posits that different plume branches may sample various regions within the LLSVP, each characterised by varying initial concentrations of primordial and recycled materials. Additionally, complex melting processes in the transition zone (Fig. 9) could involve denser material sinking while the buoyant mantle rises, leading to intricate geochemical interactions[88].

In summary, our study integrates diverse observations into a model linking asymmetric EPR accretion and surface features to the motion and buoyancy of deep Pacific LLSVP plumes. Tomography reveals a vast, deeper (below 600 km) plume mantle extending westward from the LLSVP, characterised by cooler temperatures and lower buoyancy. We link the EPR's westward migration (~92 mm/yr) to a persistent westward force exerted by the deeper, less buoyant plume mantle for at least the past 26 Ma. The Easter plume exhibits increased buoyancy above 600 km, concentrated on the Nazca Plate side of the EPR. This localised upper mantle plume buoyancy coincides with nearly double the amount of crust accreting to the Nazca Plate and the migration of the EPR away from the hotspot. This excess accretion primarily occurs through the formation of paleo-microplates, which undergo volcanic and geochemical overprinting as they become integrated into the SYGR. The Easter microplate exemplifies the latest cycle shaping the two-stage accretion observed in the SYGR's magmatic and geochemical evolution.

Finally, by correlating asymmetric EPR accretion with plume mantle movement and buoyancy, we highlight the interplay between mantle dynamics, lithospheric processes, and intraplate magmatism, crucial for shaping long-term oceanic plate boundary dynamics. Our findings suggest plumes may capture spreading ridges, contributing to an ongoing debate. The observed correlation between plume motion and the EPR underscores the importance of studying interactions between spreading ridges and the deeper, cooler, less buoyant plume mantle extending beyond them. Furthermore, the two-stage asymmetric accretion of the SYGR, linked to plume motion and increased upper mantle buoyancy, advances our understanding of hotspot track compositions, such as bilateral asymmetry. Additionally, the long-term synchronous movement of the regional EPR and the deeper, cooler Easter plume offers constraints on plume motion for absolute plate motion models.

## Methods

### $^{40}Ar/^{39}Ar$ dating

As previously described in 2019[89] and[90] the groundmass samples were prepared following the methods of ref. 91. The 200–180 μm samples measured at Oregon State were cleaned in a series of hour-long acid baths, progressing from 1 N HCl to 6 N HCl to 1 N HNO$_3$ to 3 N HNO$_3$, followed by a final milli-Q water bath. Each separate was picked by hand under a binocular microscope to ensure the removal of alteration, and to confirm the purity of the separate. Plagioclase separates were further leached using 5% HF for 15 min.

Groundmass samples were irradiated for 6 h in the CLICIT position at the Oregon State University TRIGA reactor. Incremental heating experiments were conducted for each sample. Irradiated samples were loaded into copper planchettes for analysis using a Thermo Scientific ARGUS-VI multi-collector mass spectrometer at the OSU Argon Geochronology Laboratory following the procedure described in ref. 92. All ages are calculated relative to Fish Canyon Tuff (FCT) sanidine with an age of $28.201 \pm 0.023$ Ma, $1\sigma$[93] and using the decay constants after ref. 94 The correction factors for neutron interference reactions at the TRIGA are $(2.70 \pm 0.17\%1\sigma) \times 10-4$ for $(^{36}Ar/^{37}Ar)Ca$, $(6.43 \pm 0.92\% 1\sigma) \times 10^{-4}$ for $(^{39}Ar/^{37}Ar)Ca$, $(1.21 \pm 0.09\%1\sigma) \times 10-2$ for $(^{38}Ar/^{39}Ar)K$ and $(6.07 \pm 9.65\%1\sigma) \times 10^{-4}$ for $(^{40}Ar/^{39}Ar)K$. Ages were calculated using the ArArCALC v2.7.0 software of Koppers[95], with errors including uncertainties on the blank corrections, irradiation constants, J-curve, collector calibrations, mass fractionation, and the decay of $^{37}Ar$ and $^{39}Ar$.

All ages are calculated using the corrected value for the original Steiger and Jäger's[96] constant for total $^{41}K$ decay to $^{41}Ar$ with a new value of $5.463 \pm 0.107 \times 10^{-10}$/yr (2$\sigma$) as reported by ref. 94 Air standard used is $^{40}Ar/^{36}Ar = 298.56 \pm 0.31$ (0.104% SD) as reported in ref. 97 For all other constants used in the age calculations, we refer to Table 2 in ref. 98 Individual J-values for each sample are calculated by parabolic extrapolation of the measured flux gradient against irradiation height and typically give 0.1–0.3% uncertainties (1$\sigma$). Incremental heating plateau ages and isochron ages are calculated as either plateau or weighted mean with $1/\sigma^2$ as the weighting factor[99] and as YORK2 least-square fits with correlated errors[100] using the ArArCALC v2.7.0 software from Koppers[95] available from the following website http://earthref.org/ArArCALC/.The quality of a $^{40}Ar/^{39}Ar$ step-heating experiment is assessed based on the following criteria: an acceptable age plateau (1) includes at least 50% of the gas released, (2) a mean square weighted deviation (MSWD) of ~1.0 and within the statistically allowed upper limit, (3) shows an inverse isochron with a $^{40}Ar/^{36}Ar$ intercept of $298.56 \pm 0.31$ (0.104% SD, ref. 97), and (5) a p-value > 5.

The most robust dates 16 unique samples based on single or replicated analyses are indicated in Supplementary Data 3 and Supplementary Data 4. One of these dates (64DS-3) has less than 50% $^{39}Ar$ (41%) so cannot be considered robust but rather an approximate age estimate. Dates for three plagioclase samples that are systematically lower than the corresponding groundmass age are rejected as these samples likely contain hydrothermally altered phenocrysts (ref. 25) consistent with plateau age suppression due to sericite alteration[101].

### Trace elements

As previously described in 2019[89] ~0.05 g of sample was accurately weighed into a Teflon beaker and digested in 1 ml 15 M HNO$_3$ and 3 ml 12 M HF for 12 h in sealed beakers on a hotplate at 80 °C. After cooling, 0.2 ml of HClO$_4$ was added to the sample, and the solution evaporated to incipient dryness at 120 °C. 2 ml of 15 M HNO$_3$ was added to the sample, and evaporated to near dryness, and this step was repeated twice before increasing the hotplate temperature to 150 °C and fuming off excess HClO$_4$. The sample was then redissolved in 4 ml 15 M HNO$_3$ and 4 ml H$_2$O, 2 drops of 12 M HF were added, and the sealed beakers were left on a hotplate at 80 °C for 12 h. The samples were then placed in an ultrasonic bath for 30 minutes, before heating at 80 °C for another 12 h. At this stage, all samples were completely in solution. The

sample solutions were then quantitatively transferred to 250 ml HDPE bottles and diluted to 200 g with MQ water to obtain a final solution of 2% HNO$_3$ + 0.002 M HF with a sample dilution factor of about 4000 and total dissolved solids of 250μg/ml. All reagents used were distilled in Teflon stills and diluted with MQ 18.2 Ohm water.

Trace element measurements were carried out at the GeoZentrum Nordbayern using a Thermo Scientific X-Series 2 quadrupole inductively coupled plasma mass spectrometer. Samples were introduced into the instrument through a Cetac Aridus 2 desolvating nebuliser system to reduce molecular interferences. An ESI SC-2 DX FAST autosampler was used to reduce washout times between samples. The instrument was tuned using a 5 ppb solution of Be, In and U; the typical sensitivity for 238U was $2 \times 106$ counts per second for a sample uptake rate of 50μl/min. The Ce/CeO ratio was typically > 4500, and thus corrections for the interference of oxides of Ba and the light rare-earth elements on Eu and Gd were unnecessary. Before each measurement session, the instrument was calibrated using multielement solutions covering the relevant concentration range. A mixed Be, Rh, In and Bi solution (30, 10, 10, 5 ppb) was mixed with the sample online and these elements were used as internal standards to correct for instrumental drift. Procedural blanks analysed during this work were negligible for all elements measured. Trace element data for the rock standard BHVO-2 measured as an unknown are given in Supplementary Data 5.

### Nd, Sr and Pb isotopic compositions

All samples (either glasses or aphyric basalts) were washed in 2 N HCl and double quartz distilled water. Clean pieces of glass or basalts were handpicked and crushed in a boron carbide mortar. The powder was leached with 0.1 N HCl for 10 minutes in an ultrasonic bath and rinsed in deionised water. Separations and analyses of Sr, Nd and Pb were performed from the same sample dissolution, according to the procedure described in ref. 102. Sr and Nd ratios were measured in dynamic mode on a TIMS Finnigan MAT 261. Results were corrected for mass fractionation to a value of 8.375209 for $^{88}Sr/^{86}Sr$ and 0.721903 for $^{146}Nd/^{144}Nd$. NBS987 and La Jolla standards during the period when the analyses were carried out had values of 0.71026 +/-3 (n = 68) and 0.511852 +/-10 (n = 34). Replicate analyses of the Pb isotope standard NBS981 give an average of 16.908 +/−3, 15.451 +/- 4, 36.579 +/−15 (n = 34) for $^{206}Pb/^{204}Pb$, $^{207}Pb/^{204}Pb$ and $^{208}Pb/^{204}Pb$. Pb isotope data were corrected to the values given in ref. 103. with a fractionation factor of 0.0011 per mass unit.

## Data availability

All the original $^{40}Ar/^{39}Ar$ and geochemical data generated in this study are available in the article Supplementary files.

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

## Acknowledgements
This project is funded by the Deutsche Forschungsgemeinschaft (DFG project RE 3020/17-1, M. R. & K. M. H.). We thank also the Bundesministerium fur Forschung und Technologie (BMBF) for funding the SO80a cruise and Chief Scientist Peter Stoffers and Captain Martin Kull and the crew of the R/V Sonne for the successful outcome of the expedition.

## Author contributions
J.M.O., M.R., and K.M.H. designed this study. M.R and K.M.H. carried out the trace element analyses; C.H. performed the Sr-Nd-Pb isotopic analyses and the $^{40}Ar/^{39}Ar$ analyses were conducted by A.A.P.K., D.P.M. & D.E.H., J.M.O. wrote the paper with inputs from all the co-authors.

## Funding

## Competing interests
The authors declare no competing interests
