## [Peer Review File · Nature Communications]

REVIEWER COMMENTS

Reviewer #1 (Remarks to the Author):

I have now completed my review of the ms “Synchronous motion of the Easter mantle plume and the East Pacific Rise” NCOMMS-23-61651.

This work is the result of extensive mapping, sampling and laboratory analysis of an important plate boundary in the eastern Pacific ocean by the PI’s. Their goal is to “see through” the filter of the entire oceanic crustal magmatic system to discern movements of long term reservoirs in the underlying mantle. This admixture of different mantle sources is the direct result of mantle convection, the history of the individual portions of the mantle involved and the motions of the overlying plates.

This work devotes considerable effort to coupling plate motions with the isotopic and trace element evidence for the manifestation of different mantle sources. Their goal is to document the location of the plume over time and its isotopic evolution. Their key findings are that the Easter plume is linked in convective space to the overlying East Pacific Rise spreading center, having caused a westward migration of the spreading center, while the plume itself has been fixed in space by the Pacific LLSVP underlying the mantle in that region.

Additionally the authors deconstruct the detailed spreading history of of the EPR, noting that the transfer of entire microplates from the Pacific to the Nazca plate has forced a westward migration of the EPR, in concert with the changes in geochemistry over the last 20Ma.

The study is a sweeping reinterpretation of the tectonic history of the eastern Pacific, incorporating isotopic, trace element and geochronologic evidence. As such it is certainly of sufficient impact to justify publication in Nature Communications. I feel that the data presented support the conclusions regarding the tectonic and geochemical evolution and I recommend its publication with changes outlined below:

I am confused by the structure of the paper. I’m used to seeing a conclusion section at the end of the paper that integrates the themes of the discussion and discusses their broader implications. Instead, each part of the discussion section seems to have its own summary paragraph. Lines 276, 383, 455, 532 and 658 each begin “In summary”, but I get to the end of the substantive part of the paper and find... a methods section? And no conclusions? This is a big paper, with a lot of different kinds of very in-depth data and analysis, but it feels like five (or more) separate papers. Restructuring to integrate the whole with its parts is badly needed here.

I recall a debate some years ago about ridge centered hotspots: whether ridges capture plumes or plumes capture ridges. It would seem from their evidence that here it’s the latter – isn’t this the sort of global process they were setting out to figure out? I couldn’t find where this was discussed in the immensity of the document.

Figures 6 and 7 are missing axis labels and units.

Figure 8 should have a north arrow, or some other indication that these data are in map view.

Reviewer #2 (Remarks to the Author):

There is really a lot in this paper! There is a lot of data and interpretations that took me some very detailed reading to properly digest. The different data threads however are drawn together in a logical fashion that together to form a cohesive model for the evolution of this entire region. I really enjoyed reading this manuscript and feel that it makes very valuable contributions.

In particular, I really like the detailed plate tectonic reconstruction and seismic tomography analysis that are used to clearly explain the formation and relationship among all the seafloor features in this region, including the apparent discrepancy in Nazca plate motions, in terms of the interaction between the plume and the evolution of the spreading ridge and microplates.

I have very few comments on the science presented in this paper and would be very happy to see it published with the text largely as is. My only real comments relate to the details of the figures, which could do with some tidying up around font sizes, labelling, colours, and consistency between the figure and the caption. These are detailed below along with a couple of minor points on the text.

All figures: many/most labels are all too small to read easily, even when zooming in on the pdf. Admittedly I'm getting on and likely need reading glasses but I could not read the figure labels in the version I printed and had to zoom a lot on screen to solve the problem.

Figure 3: what are the values ~ 64.6 (in red) and 191.6 (in blue) referring to? Are these also mm/yr? The caption text refers to small open symbols but I cannot see these. Are the small orange squares referred to as small red boxes in the caption? If yes, then there's a mismatch between part A where these are red dots and part B where they are orange squares. There are three different red symbols used, might be easier with additional colours? Also, might be good to have exactly the same colours and symbols for all the features between part A and B of this figure?

Figure 4: Is the grey rectangle shading the VER1 sites? If yes, maybe move the label? (better in Figure 5, which is where I figured it out).

Figure 6: I find it disconcerting that part B above part A. Missing axes labels on part B.

Figure 9: cannot see red disks? Orange maybe?

Figure 10: Would be great to have the green dots that are on the cut-through views also on the map view. Could the plate boundaries shown on the map view seismic velocity maps also be shown on the map view for easy reference. $19-12^{\circ}\text{S}$ mentioned in text but latitudes unreadable on map view even very zoomed in.

Line 50-51: Cannot see Easter Island on Figure 1. Add reference to Figure 2 also. Or mark on Fig 1.

Line 50 vs Line 64: Two slightly different suggested locations for the Easter plume. Might be good to clarify where these come from.

Line 92: briefly detail the SO voyage?

Line 258: shifting position of Easter Island. I think it's the shifting position of Easter Island relative to the Easter plume. Maybe remind readers here that Easter Island has ages from 3Ma?

Line 261: it would be good to see this working out somewhere.

Line 266: Where did the 170 km come from?

Line 430 – maybe this could be a separate section?

Line 547: Title of this section is a bit confusing as I can't find the two-stage model being already introduced explicitly. It is briefly mentioned on line 451, but maybe a quick (re-) introduction here would be handy.

Point-by-point response to the reviewers' comments, reproduced verbatim

Responses to reviewers

Reviewer #1 (Remarks to the Author):

This work is the result of extensive mapping, sampling and laboratory analysis of an important plate boundary in the eastern Pacific ocean by the PI's. Their goal is to "see through" the filter of the entire oceanic crustal magmatic system to discern movements of long term reservoirs in the underlying mantle. This admixture of different mantle sources is the direct result of mantle convection, the history of the individual portions of the mantle involved and the motions of the overlying plates.

This work devotes considerable effort to coupling plate motions with the isotopic and trace element evidence for the manifestation of different mantle sources. Their goal is to document the location of the plume over time and its isotopic evolution. Their key findings are that the Easter plume is linked in convective space to the overlying East Pacific Rise spreading center, having caused a westward migration of the spreading center, while the plume itself has been fixed in space by the Pacific LLSVP underlying the mantle in that region.

Additionally the authors deconstruct the detailed spreading history of the EPR, noting that the transfer of entire microplates from the Pacific to the Nazca plate has forced a westward migration of the EPR, in concert with the changes in geochemistry over the last 20Ma.

The study is a sweeping reinterpretation of the tectonic history of the eastern Pacific, incorporating isotopic, trace element and geochronologic evidence. As such it is certainly of sufficient impact to justify publication in Nature Communications. I feel that the data presented support the conclusions regarding the tectonic and geochemical evolution and I recommend its publication with changes outlined below:

We greatly appreciate the reviewer's insightful summary of our work. We have carefully incorporated his feedback into our revised manuscript to enhance both clarity and depth. This integration has not only enhanced our discussion but also significantly strengthened our conclusions.

I am confused by the structure of the paper. I'm used to seeing a conclusion section at the end of the paper that integrates the themes of the discussion and discusses their broader implications. Instead, each part of the discussion section seems to have its own summary paragraph. Lines 276, 383, 455, 532 and 658 each begin "In summary", but I get to the end of the substantive part of the paper and find... a methods section? And no conclusions? This is a big paper, with a lot of different kinds of very in-depth data and analysis, but it feels like five (or more) separate papers. Restructuring to integrate the whole with its parts is badly needed here. We strongly agree with the reviewer's suggestion regarding the need for better integration of the various sections of our manuscript. Recognizing the complexity of the content, we have undertaken substantial revisions to enhance the coherence and flow throughout.

Specifically, we have revised the abstract and made significant additions across the manuscript, particularly in the introduction, to ensure a cohesive presentation of our research. Additionally, we

have introduced a new 'Summary & Conclusions' section. These revisions aim to seamlessly integrate the various parts of the manuscript and provide a clear, continuous narrative that is closely linked to the data presented.

We believe these revisions much improve the manuscript's readability and effectively convey the significance of our findings.

I recall a debate some years ago about ridge centered hotspots: whether ridges capture plumes or plumes capture ridges. It would seem from their evidence that here it's the latter – isn't this the sort of global process they were setting out to figure out? I couldn't find where this was discussed in the immensity of the document.

We appreciate the reviewer highlighting this crucial aspect of our study. In response, we have expanded on this point in the final paragraph of the revised manuscript to ensure that it is prominently addressed within the broader implications of our work. We also mention now in the abstract that the plume controls the location of the EPR.

Figures 6 and 7 are missing axis labels and units.

Fixed

Figure 8 should have a north arrow, or some other indication that these data are in map view.

Done

Reviewer #2 (Remarks to the Author):

There is really a lot in this paper! There is a lot of data and interpretations that took me some very detailed reading to properly digest. The different data threads however are drawn together in a logical fashion that together to form a cohesive model for the evolution of this entire region. I really enjoyed reading this manuscript and feel that it makes very valuable contributions.

In particular, I really like the detailed plate tectonic reconstruction and seismic tomography analysis that are used to clearly explain the formation and relationship among all the seafloor features in this region, including the apparent discrepancy in Nazca plate motions, in terms of the interaction between the plume and the evolution of the spreading ridge and microplates.

We are very grateful for this reviewer's insightful summary of our work, which has helped us to clarify and enhance our presentation.

I have very few comments on the science presented in this paper and would be very happy to see it published with the text largely as is. My only real comments relate to the details of the figures, which could do with some tidying up around font sizes, labelling, colours, and consistency between the figure and the caption. These are detailed below along with a couple of minor points on the text.

The reviewer's comments have proven invaluable in enhancing the clarity and effectiveness of our figures, which now better support the arguments underpinning our conclusions. We have made significant modifications to font sizes, labeling, color schemes, and consistency between the figures and their captions along with adding more labels. Additionally, we have incorporated changes to the text as suggested.

All figures: many/most labels are all too small to read easily, even when zooming in on the pdf. Admittedly I'm getting on and likely need reading glasses but I could not read the figure labels in the version I printed and had to zoom a lot on screen to solve the problem.

We have enlarged the figure labels as much as possible to enhance readability. Additionally, should our manuscript be accepted, we will ask during the production phase that the final figures are as large as possible to further optimize their impact and clarity for readers.

Figure 3:

what are the values ~64.6 (in red) and 191.6 (in blue) referring to? Are these also mm/yr?

We have added mm/yr to these numbers on the figure and revised the caption as follows:

"The Pacific trend (64.6 mm/yr; solid red line and diamonds...."

"The microplate trend (191.6 mm/yr; diagonal solid blue line)..."

The caption text refers to small open symbols but I cannot see these.

Thank you for identifying this error. We have removed the sentence in question from the revised text.

Are the small orange squares referred to as small red boxes in the caption? If yes, then there's a mismatch between part A where these are red dots and part B where they are orange squares.

Thank you for your observation regarding the small orange squares. In the revised version, we have changed these indicators to yellow, which also aligns better with the Easter-type isotopic compositions shown in Figure 5.

There are three different red symbols used, might be easier with additional colours? Also, might be good to have exactly the same colours and symbols for all the features between part A and B of this figure?

Very good point. Here is the revised section of the relevant section of the caption.

"The small yellow boxes represent published $^{40}\text{Ar}/^{39}\text{Ar}$ dates for submarine volcanism at Umu, Moai, and Easter Island (O'Connor et al., 1995)."

Figure 4: Is the grey rectangle shading the VER1 sites? If yes, maybe move the label? (better in Figure 5, which is where I figured it out).

Done. We have also added an arrow for further clarity

Figure 6: I find it disconcerting that part B is above part A. Missing axes labels on part B.

Part A is now above part B and labels have been added to the axes labels (also in Figure 7). We have also added location labels to the map view to improve clarity.

Figure 9: cannot see red disks? Orange maybe?

We've clarified in the caption that the large red disks are shown with 42% transparency to show underlying seafloor structure.

Figure 10:

Would be great to have the green dots that are on the cut-through views also on the map view.

Done

Could the plate boundaries shown on the map view seismic velocity maps also be shown on the map view for easy reference.

Done

19-12°S mentioned in text but latitudes unreadable on map view even very zoomed in.

Fixed

Line 50-51: Cannot see Easter Island on Figure 1. Add reference to Figure 2 also. Or mark on Fig 1.

Fixed

Line 50 vs Line 64: Two slightly different suggested locations for the Easter plume. Might be good to clarify where these come from.

Very good point. We have revised the relevant text to read as follows:

“Notably, the Easter Island (aka Rapa Nui) region, situated east of the Easter Microplate on the Nazca Plate and adjacent to the East Pacific Rise (EPR) (Figs. 1 & 2), is linked to a deep, lower mantle plume origin (e.g., Courtillot et al. 2003; Montelli et al., 2006; French & Romanowicz, 2015; Hosseini et al., 2020; Koppers et al. 2021).”

“Published $^{40}\text{Ar}/^{39}\text{Ar}$ dates for basaltic rock samples dredged from the Salas y Gómez Ridge (SYGR) and the Nazca Ridge are consistent with age-progressive volcanism currently originating from near Salas y Gómez Island (approximately 106°W) (Ray et al., 2012; O’Connor et al., 1995).”

Line 92: briefly detail the SO voyage?

We have added the following to the revised text:

“During the RV SONNE 80a-Midplate III expedition (Valparaiso - Easter Island, Stoffers et al., 1992)...”

Stoffers et al, 1992 refers to the cruise report, which is available online.

Line 258: shifting position of Easter Island. I think it’s the shifting position of Easter Island relative to the Easter plume.

The revised sentence now reads as follows:

“We have also employed magnetic anomaly data for the Easter microplate (Fig. 7) to establish a correlation between the region-wide synchronous westward (~ 92 mm/yr) movement of the EPR and the Easter plume and the evolution of the Easter microplate (Fig. 7).”

Maybe remind readers here that Easter Island has ages from 3Ma?

Looking again at the Vezzoli & Acocella (2009) compilation of K-Ar ages for Easter Island we have added the following text to the Figure 3 caption:

" Additionally, a compilation of published K-Ar ages indicates that the three shield volcanoes on Easter Island—Poike, Rano Kau, and Terevaka—underwent two phases of activity between 0.78

and 0.3 Ma and 0.24 and 0.11 Ma (Vezzoli & Acocella (2009). These phases are marked by vertical black lines. Three K-Ar ages ranging from 3 to 1.89 Ma are considered unreliable, as discussed in Vezzoli & Acocella (2009)."

Line 261: it would be good to see this working out somewhere.

Line 266: Where did the 170 km come from?

We have address both these issues by revising the main text as follows:

“A comparison of $^{40}\text{Ar}/^{39}\text{Ar}$ isotopic dates with seafloor anomaly ages (Fig. 6) indicates that the ‘Pacific’ trend lavas erupted approximately 170 km west of the nearest spreading axis, with the underlying crust being around 2 Ma at the time of eruption.”

Furthermore, we have added text to the Figure 6 caption as follows:

“The $^{40}\text{Ar}/^{39}\text{Ar}$ dates defining the ‘Pacific’ trend (red diamonds and black dashed line) align with the migration of the EPR (West Rift) crust (circle symbols and black regression line). The distance between these trends implies that the ‘Pacific’ trend volcanism erupted about 170 km west of the EPR (West Rift).”

Line 430 – maybe this could be a separate section?

We appreciate the valuable suggestion. We have introduced a new and expanded section titled "Absolute Plate Motion" to further clarify and focus on this important aspect of our study.

Line 547: Title of this section is a bit confusing as I can't find the two-stage model being already introduced explicitly. It is briefly mentioned on line 451, but maybe a quick (re-) introduction here would be handy.

Thank you for the suggestion. We have incorporated a brief reintroduction at the beginning of the section to provide readers with a concise overview, enhancing their understanding of the subsequent content.

REVIEWERS' COMMENTS

Reviewer #2 (Remarks to the Author):

I raised some concerns about the presentation of some figures and working out in the previous version of this manuscript. I feel that these concerns have been addressed in this revised version of the manuscript and have no concerns about the the data and ability to support conclusions.

In addition, the text changes undertaken on the advice of the other reviewer has also improved the paper.

Very happy to see this published.

Point-by-point response to the reviewer 2's comments on Version 2

I raised some concerns about the presentation of some figures and working out in the previous version of this manuscript. I feel that these concerns have been addressed in this revised version of the manuscript and have no concerns about the data and ability to support conclusions.

In addition, the text changes undertaken on the advice of the other reviewer has also improved the paper.

Very happy to see this published.

We greatly appreciate the reviewer's help in improving the presentation of the figures and refining this revised version of the manuscript.